# Sharing Data and Transferring Samples Within Pediatric Clinical Studies: How to Overcome Challenges and Make Them a Science Opportunity

**DOI:** 10.3390/healthcare12232473

**Published:** 2024-12-06

**Authors:** Annalisa Landi, Federica D’Ambrosio, Silvia Faggion, Francesca Rocchi, Carla Paganin, Maria Grazia Lain, Adriana Ceci, Viviana Giannuzzi

**Affiliations:** 1Fondazione per la Ricerca Farmacologica Gianni Benzi Onlus, 70124 Bari, Italy; al@benzifoundation.org (A.L.); adriceci.uni@gmail.com (A.C.); 2Fondazione Penta ETS, 35127 Padova, Italy; federica.dambrosio@pentafoundation.org (F.D.); silvia.faggion@pentafoundation.org (S.F.); 3Ospedale Pediatrico Bambino Gesù, 00152 Roma, Italy; francesca.rocchi@opbg.net (F.R.); carla.paganin@opbg.net (C.P.); 4Fundação Ariel Contra o SIDA Pediátrico, Maputo P.O. Box 2822, Mozambique; mlain@arielglaser.org.mz

**Keywords:** pediatric clinical studies, transfer of samples, data sharing, regulatory, ethics

## Abstract

EPIICAL (Early treated Perinatally HIV-Infected individuals: Improving Children’s Actual Life) is a consortium of European and non-European research-driven organizations inter-connected with the aim of establishing a clinical and experimental platform for the early identification of novel therapeutic strategies for the pediatric Human Immunodeficiency Virus (HIV). Within the EPIICAL project, several pediatric clinical studies were conducted, requiring the collection and transfer of biological samples and associated data across boundaries within and outside Europe. To ensure compliance with the applicable rules on pediatric data and sample transfer and to support the efforts of academic partners, which may not always have the necessary expertise and resources in place for designing, managing and conducting multi-national studies, the consortium established a dedicated expert Working Group. This group has guided the consortium since the start of the project through the complexities of the ethical and regulatory aspects of international clinical studies. The group provided support in the design and preparation of the prospective and retrospective multi-center and multi-national pediatric studies with a focus on the clinical study protocols, informed consent and assent forms. In particular, well-structured informed consent and assent templates were developed, and data sharing and material transfer agreements were set up to regulate the transfer of samples among partners and sites. We considered that such support and the implementation of ad hoc agreements could provide effective practical solutions for addressing ethical and regulatory hurdles related to sharing data and transferring samples in international pediatric clinical research.

## 1. Introduction

Early treated Perinatally HIV-Infected individuals: Improving Children’s Actual Life —EPIICAL [1]—is a consortium of European Union (EU) and non-European (non-EU) research-driven organizations and academic institutions aiming at implementing a predictive platform for the early identification of novel therapeutic strategies for children affected by the human immunodeficiency virus (HIV). It foresaw the design and conduct of studies in pediatric HIV populations in Europe, Africa and Asia. This involves developing and applying statistical and mathematical modeling to data derived from cohorts of early treated infants and children to identify virological, immunological and transcriptomic profiles associated with early control of HIV infection after antiretroviral therapy (ART) initiation as well as viral control following ART interruption.

EPIICAL consists of a large number of partners from all over the world and has been running for eight years.

The foresight of such a consortium was to anticipate potential ethical and regulatory issues related to the planned pediatric studies and to involve experts in the field from the outset of activities. Therefore, in the framework of the EPIICAL project, a Working Group (WG) with ethics and regulatory experts was set up to ensure that all relevant rules are complied with.

This paper aims to describe our experience coming from the EPIICAL project as an extensive work aimed to investigate and address the challenges related to the transfer of samples and associated data across boundaries within and outside Europe in the context of pediatric clinical studies. Possible solutions to overcome them will be emphasized as well. These would result in useful tools and strategies for other researchers working in different disease areas.

## 2. Setup of the Activities

Biological samples, like blood, tissue, urine and saliva are commonly used in biomedical research and their analyses provide key outputs in clinical studies. Regulatory, legal and ethical considerations, including but not limited to informed consent, assent from minors and data protection, particularly with respect to long-term storage of samples and related data, must be taken into account [2,3].

Several challenges can be identified when dealing with the transfer of samples and associated data across boundaries in the context of clinical studies. Among them, ethical challenges relate to the privacy of individuals and data control [4] and the respect for informed consent; furthermore, the regulatory challenges associated with the application of national provisions ruling the transfer of samples and associated data make the situation even more complex.

Such challenges are emphasized when vulnerable subjects, such as minors, are involved [3].

International clinical studies might represent a further complication since the ethics, regulatory and data protection framework regarding the sharing of samples and associated data for scientific purposes seems scattered among EU and non-EU countries [5]. In fact, countries have different laws regarding the use of clinical samples, especially when dealing with children. In addition, language barriers further complicate the conduct of multi-national studies.

The implementation of multi-national collaborative projects with a focus on data and sample sharing often faces regulatory roadblocks that slow progress. This has been exacerbated by the entry into force of the European General Data Protection Regulation Reg (EU) 2016/679 (GDPR) [6], which, by leaving a significant part of decision-making to the Member States, has led to confusion and bureaucratic complexity, particularly when non-EU partners are involved [7]. Although there is a lack of harmonized frameworks or guidelines across the world, there are many strategies that might be implemented to address the challenges outlined above. First of all, the transparency of the information to be provided to the study subjects and/or to their parents/legally designated representatives regarding the transfer and use/future use of samples and associated data to achieve the study purposes (e.g., analysis in specialized laboratories) is needed. This information shall always be included in the study protocol with a description of how data and samples are processed, as well as in the informed consent form of the study participants or their parents/legally designated representatives. When seeking consent, the use, storage, and possible future use of the material should also be explained [3]. Moreover, when dealing with minors, children should participate in the informed consent and assent process according to their age and understanding and receive age-appropriate information about what will happen in the study as well [3]. Finally, we considered that ad hoc agreements regulating the sharing of samples and associated data shall be set up to ensure the lawful sharing of data and samples among sites and countries. These agreements constitute mechanisms to ensure uniformity of data and sample access across projects and countries and may be regarded as consistent basic agreements for addressing data and material sharing globally [8].

The dedicated WG supported the investigators in the relevant ethical and regulatory applications during the whole duration of the clinical studies and deemed it necessary to involve the Sponsor’s representatives in the group from the beginning. It started its activities by providing support in the design and preparation of the EPIICAL prospective and retrospective multi-center and multi-national pediatric studies. They involved both EU and non-EU countries: South Africa, Mozambique, Mali, Uganda, Thailand, Italy, the United Kingdom, Spain and the United States.

The group followed a centralized approach, ensuring uniform ethical standards across all countries and sites, as described below:-The applicable regulatory and ethics provisions were identified through an analysis of the international framework: the Helsinki Declaration for Ethics in Human Subjects (2013) [9]; International Ethical Guidelines for Health-related Research Involving Humans CIOMS-WHO (2016) [10]; Additional Protocol to the Oviedo Convention on biomedical research (2005) [11]; European Commission Ethical considerations for clinical trials on medicinal products conducted with minors (2017) [12]; and European General Data Protection Regulation Reg (EU) 2016/679 (GDPR) [6]. They were considered to complement the national frameworks. In particular, the 2017 EC considerations for clinical trials in minors [12] were followed to verify the limit of blood amount to be taken from minors, while the CIOMS WHO guidelines [10] were followed for implementing separate consent for genetic testing.-A core package of documents was prepared to submit the studies to the competent Ethics Committees in all clinical sites involved in compliance with the national rules and international standards.-The clinical study protocol was reviewed and any necessary amendments were made to align it with local requirements and to develop and release a unique version for all sites.-Data and sample flows were identified for all the EPIICAL studies to better illustrate the ethics and regulatory needs, including those specifically related to the transfer of health and genetic data with non-EU countries.-Informed consent and assent templates were prepared to be adapted to local requirements, and support was given to implement data protection and confidentiality rules. Considering that the studies foresaw the transfer of samples and associated data, information on the transfer was provided to study participants in the parent information sheet and informed consent forms for minors.-A Standard Operating Procedure was released on the management of personal data and samples and on consent requirements in 2018. Then, once GDPR [6] entered into force, a letter was prepared for the investigators to help them fully comply with the new EU privacy legislation and implement data protection and confidentiality rules.

## 3. Informed Consent and Assent

We deemed a well-structured informed consent template as the most suitable solution to address the national differences and then overcome the related challenges in samples and data sharing. For this reason, in the framework of the EPIICAL project, a parent information sheet and informed consent form template were prepared (available as Appendix A). Given the pediatric specificities of the EPIICAL studies, the information sheet and informed consent form were addressed to the parents/legally designated representatives of the minor patients.

The information sheet included all the relevant information on the study and the use of samples and associated data, including information about the transfer; in particular, it included the following:-The type of samples and data to be collected and shared.-The purposes of the transfer (i.e., analysis in specialized laboratories).-The rights of children, including the subject’s confidentiality and all the rights under GDPR.-The commitment of the Sponsor to ensure compliance with the applicable data protection rules.-The explanation of the adoption of de-identification measures to protect patients’ privacy.-The storage location and duration of the data and samples and the countries/cities where the laboratories are located.-The future use of remaining samples and the availability to be recontacted for possible further testing and then refreshing consent.

Study participants were also reassured that the Ethics Committee(s) approved the study and would approve any possible modification to the study, e.g., transfers to other locations not defined yet at the start of the study and new tests on the remaining samples.

In order to apply the minimization principle (as stated in the GDPR [6]), study participants were informed that only information essential for the purposes of the study would be collected.

Patients and their parents/legal guardians were informed of their right to withdraw at any time and without giving a reason for the decision, including the destruction of the remaining samples and associated data, unless already analyzed.

A granular consent section was foreseen to allow participants to make some choices related to the use of their samples and associated data. These choices included the transfer of samples to specific laboratories for analysis, the re-use of remaining samples and associated data and the performance of genetic testing. With reference to this latter point, participants shall be informed about any possible unexpected or incidental findings coming out from the research and if any treatment or preventive measures are available. This information was collected. Parents/legal guardians were also given the chance to refuse the inclusion of their children’s data in an aggregated and anonymized form within reports/articles prepared for dissemination and communication purposes. The possibility to be re-contacted for further research studies was added as well.

Finally, in the event that the parent/legal guardian was unable to read and sign the parent information sheet and the informed consent form, the consent process implemented in EPIICAL foresaw (1) the involvement of a person in charge of reading and explaining the contents of the information sheet and (2) the thumbprint of the parent/legal guardian accompanied by the name and signature of a witness, defined as “a person independent from the research team or any team member and who was not involved in obtaining consent” to provide written consent. This provision was sourced from the EU Clinical Trials Regulation (CTR) [13], which rules interventional clinical trials in the EU.

Furthermore, considering the nature of EPIICAL studies, assent form templates were prepared for children and adolescents as well (available as Appendix A). Plain and clear language was used to provide information to children according to their age and maturity. The study procedures and transfer of samples to other laboratories were explained.

Children were informed that they could choose not to take part in the study or to change their minds at any time without providing a reason. The basic concept of privacy was also included in the form to confirm that their identity shall be kept secret. Finally, they were informed that when they reach the age of maturity, they will have the possibility to re-evaluate their participation and to consent or object to any further use of their data.

## 4. Data Sharing Agreement

Clinical data from EPIICAL studies were entered by the study team at each participating center into a centralized database provided by the Sponsor (REDCap).

Each site was responsible for entering the data collected at their site, with access restricted to the dataset pertaining to their own patients. Access to REDCap was granted only to the clinical site’s staff trained in the study protocol and the use of the database. Access was provided via email with a username and a temporary password, to be replaced before their first login, adhering to the security criteria established by the Sponsor.

REDCap automatically prompted the user to update the password every ninety days.

Each patient enrolled in the study was assigned an alphanumeric code based on the principle of pseudonymization.

As part of the project activities (labs, center for statistical analysis, etc.), access to the database was also granted to other consortium partners following the same procedures outlined above. Depending on the delegated tasks, each partner was given access only to the dataset necessary to perform their specific activities as required by the study protocols.

A Data Sharing Agreement (DSA) was prepared for each EPIICAL study to regulate the data flow, as required by the GDPR [6]. The DSA focuses on EU privacy legislation since the Sponsor and some parties involved in the study were based in the EU, which means that the GDPR applies. Moreover, since the Sponsor is responsible for the conduct of the study and the activities delegated to the institutions involved in it, the Sponsor had to ensure compliance with the GDPR for all processing activities, regardless of whether they were carried out in the EU or outside the EU.

As a preliminary step, the study data flows were mapped, with parties being either senders/exporters or recipients/importers based on the role of each party in the study according to the protocol (clinical site, laboratory, chief investigator, etc.) and on the type of personal data being processed.

Following that, the DSA allocates the privacy roles of the parties according to each party’s contribution to the study, defines the study processing activities’ details (nature and purpose(s), categories of data subjects, categories of the data processed, frequency of transfer, data retention period) and sets out the obligations the parties are subject to when performing the personal data processing activities necessary to conduct the study. Such obligations include the responsibility of the parties to implement and maintain technical and organizational measures to ensure the security of personal data and the protection of data subjects’ rights and freedoms.

Importantly, the DSA also regulates the transfer of personal data to third countries, ensuring that any transfer of personal data to a third country or an international organization within the study takes place in compliance with the GDPR.

Last but not least, on the assumption that by being based outside of the EU, many parties may not be familiar with the EU privacy legislation, some training materials were provided as an annex to the DSA, with the aim of helping parties get a better understanding of the basic concepts of the GDPR and thus of their obligations under the DSA.

## 5. Material Transfer Agreements

The setup of Material Transfer Agreements (MTAs) was considered relevant in the framework of the multi-national transfer of samples and associated data when conducting pediatric clinical studies, especially if it involves non-EU countries.

Therefore, for the EPIICAL studies, MTA templates were developed starting from the model provided by Mascalzoni et al. [8] to regulate the transfer of human samples among partners and concerned clinical sites (available as Appendix A). They complemented the research project collaboration agreements, and the above-mentioned DSA set up by the Sponsor.

Two main figures were identified for each site transferring samples and associated data: the provider, the registered legal entity in charge of providing biospecimens and associated data to the recipient, and the recipient, the registered legal entity in charge of receiving biospecimens and associated data from the provider.

Information about the Sponsor of the study and the coordinating and principal investigators was included, as well as the definitions of specific terms (e.g., provider, recipient, informed consent, etc.).

The agreement included information and declarations from the provider and from the recipient.

The provider is required to do the following:-Confirm the alignment of the regulatory and ethical framework of the country concerned with the international provisions concerning medical research.-Guarantee research quality, security and privacy protection.-Confirm that the international transfer of biospecimens and personal data is allowed.-Ensure compliance with the international quality standards for human biological materials and the specific methods/measures applicable to the type of biospecimens.-Describe the legal basis for the storage and distribution and for allowing biospecimens and data sharing by stating that informed consent was obtained from subjects or their parents/legal representatives in case of minors. It was accompanied by the informed assent, where required.-Include information on the expected number of individuals providing data and samples as well as on the type of data (e.g., outcomes of clinical/laboratory/instrumental analyses, medical records, genetic testing results, Case Report Forms) and samples (e.g., blood samples, tissue type, cell preparation, DNA, RNA, protein)-Describe the applied de-identification measures as well as information on the storage location and the modalities of transfer. Details about shipping and the applicable regulations (e.g., the International Air Transport Association, the European Agreement on International Carriage of Dangerous Goods) are requested.-Define what happens at the end of the agreement with the shared data and samples. Two options are proposed: to destroy or return them to the provider. A written notification/certification with the confirmation of the destruction/anonymization is mandatory at the end of the agreement.

The recipient is required to do the following:-Confirm compliance with applicable regulations, policies and guidelines, as well as with the study protocol.-Declare that data and samples will be used only for the purposes established in the agreement and in the framework of the EPIICAL project and that they will not be transferred to other facilities or institutions without written consent from the provider.-Undertake not to use data and samples in case of withdrawn consent and then destroy or return them.

Import and export licenses were also necessary, when requested by national laws, to make the transfer across boundaries of biospecimens lawful.

## 6. Discussion and Conclusions

The EPIICAL experience highlights well-known challenges related to the transfer of samples and associated data within pediatric clinical studies and reveals solutions proposed by the authors on how to overcome them and make them a scientific opportunity for researchers and healthcare professionals.

Firstly, we believe that the involvement of ethics and regulatory experts from the study design stage is relevant and valuable in supporting the design and conduct of studies and providing continuous advice to the study team. It allowed for the smoother execution of study activities to set up and conduct multi-national studies.

Such an involvement is intended to ensure that all study activities are performed according to the applicable rules and ethical standards, reduce differences and inequities across countries from different political and economic settings. This could be particularly challenging for academic partners, who may not always have the necessary expertise and resources in place for designing and managing multi-national studies [5,14]. Therefore, public–private collaborations or collaborations among different stakeholders, possibly in the framework of research funds, would be crucial for the future.

We emphasized that clear and easily accessible information related to the transfer of samples and associated data in the context of clinical studies must be provided in the informed consent and assent forms and that MTA and DSA represent useful means to regulate the transfer of samples among countries and institutions as well as to regulate the data flow, as required by the GDPR. Of course, setting up these agreements was not an easy task to do. This was mainly due to the lack of common internationally agreed rules and a consequent lack of harmonization of the regulatory framework across countries involved in multi-national clinical studies. In fact, such agreements were set up considering and incorporating all the applicable legislation.

A homogeneous support group aimed at guiding and monitoring the research can reduce differences and inequalities and ensure transparency in human research and ethical principles, including scientific partners from countries considered both rich and poor. Furthermore, a well-structured submission package streamlines the ethical submission process and reduces the time required for ethics committees to approve the study. Additionally, awareness of regulatory procedures might facilitate the sharing of samples and associated data.

Table 1 provides an overview of the ethics and regulatory issues that arose during our pediatric research project in the geographical areas involved and how they were addressed through the careful regulatory frameworks established by the consortium. For example, we consider the preparation of a letter for investigators, with practical information on how to comply with GDPR (e.g., how to modify the informed consent and assent processes and documents), as one of the “success stories” of this work. This is because we shared such a letter even before the full application of the ‘new rule’, i.e., GDPR. Another example is the agreement that was reached among clinicians and regulatory experts on the type of clinical studies foreseen in the project. Discussion and consultation of relevant applicable documents were adopted to solve the challenge.

Our experience has highlighted that real global harmonization of multi-national, multi-continental clinical studies not investigating any medicine is difficult to reach without a global regulatory framework like the ICH, and this becomes even more relevant in the light of the growing use of multi-sources data (e.g., studies involving primary and secondary data sources). Harmonization should also be pursued considering the international dimension of scientific research [7].

As a future direction, close collaboration and efficient communication between the study team and the regulatory/ethics experts should be pursued, achieved and maintained for the whole duration of the clinical studies. We consider this as a relevant action, considering that ethics and regulatory activities need timely planning (e.g., obtaining ethics approval, protocol amendments, etc.), especially in case of new ideas or changes to the original plans. Furthermore, a set of expected outcomes and key performance indicators could be identified and measured during a multi-national clinical study [15] to value and monitor the work done from an ethics/regulatory WG and to promptly identify issues and related solutions.

EXPECTED OUTCOMES:-To accelerate the ethics approval and/or competent authority authorization;-To reduce the number of requests for modifications/integrations from ethics committees and competent authorities;-To speed up the start of data and sample sharing and, therefore, of their analysis.

KEY PERFORMANCE INDICATORS:-Number of periodical group meetings, number of requests for support received from Sponsor or investigators and time to resolve a request for ethics/regulatory support;-Number of requests for clarification/document modifications received by ethics committees and/or competent authorities out of the number of applications;-Time for agreeing DSAs and MTAs.

These metrics, as well as others related to the execution phases of a pediatric clinical study (set up, enrolment, conduct, etc.), should be regularly measured and published.

The best possible guidance on how to deal with any ethical or regulatory issues that may arise during the study can only be provided if the regulatory team is promptly informed about them.

This work aimed to provide the scientific community with some learnings on how to build a strong consortium in the framework of pediatric studies and how to manage the ethics and regulatory issues related to pediatric samples and data sharing. This can also help to speed up study procedures and strategies.

## Figures and Tables

**Table 1 healthcare-12-02473-t001:** Ethics and regulatory issues arose during the pediatric research project in the geographical areas involved.

Continent	Ethics/Regulatory Issues	Solutions
America	Acceptance of EU standards for clinical research, data protection and confidentiality	In the agreements: To refer to the international standards, e.g., ICH, “as implemented in the national legislation”. To specify that provisions apply “to the extent the EU rule is compatible with the national laws”.
Africa	Storage of samples abroad for future studies not permitted.	Biospecimens for future studies only stored locally at the clinical sites.
America	Need to comply with local laws and requirements.	The researchers were asked to comply with both EU and local laws.
Africa, Europe	Divergent classification of the clinical study (observational, non-interventional, non-pharmacological, etc.).	Upfront agreed classification among clinicians and regulatory experts on the type of clinical studies foreseen in the project, i.e., non-pharmacological clinical study.
Africa, Europe	Need to limit blood withdrawals from children according to their age and weight.	Agreement among clinicians and regulatory experts to follow European ethical recommendations on pediatric studies regarding blood withdrawals from children.
Africa, Europe	Need to transfer samples outside the country in compliance with applicable laws.	To set up regulatory-sounded Material Transfer Agreements for sharing samples.
Europe	Results of the studies mandatorily shared with parents/legal representatives.	Procedure specified in the informed consent process: parents/legal representatives informed about their right to receive study results in the informed consent document.
Europe	Future uses of samples and related data not to be broad but related to the original study and approved by an ethics committee.	Future uses of samples and data specified in protocol and informed consent documents, as approved by an ethics committee.
Europe	Informed consent more user-friendly language, limiting medico-legal terminology wherever possible.	Informed consent and assent documents adapted to use user-friendly language, minimizing medico-legal terminology and revised by ELSI experts.
Europe	Need to use user-friendly language in informed assent documents.	
Africa, America, Europe	Need to update EU laws because of modifications, e.g., Directive 95/46/EC repealed by GDPR.	Investigators provided with practical information on how to comply with GDPR requirements on informed consent and assent process and documents, in particular with those not already included in the studies.

## Data Availability

No generated data.

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
