# Peer review of "Sharing Data and Transferring Samples Within Pediatric Clinical Studies: How to Overcome Challenges and Make Them a Science Opportunity"

_healthcare, 2024, doi:10.3390/healthcare12232473_

Round 1
Reviewer 1 Report
Comments and Suggestions for Authors
The authors achieved by their actions addressing several steps and topics there are necessary for the use of scientific research. And they were able to bild a unified project for different countries, which have their own rules and ethical conditions

Author Response
COMMENTS 1:
The authors achieved by their actions addressing several steps and topics there are necessary for the use of scientific research. And they were able to bild a unified project for different countries, which have their own rules and ethical conditions.
The main issue of this article was to present the experience of a clinical and experimental platform that could help the development of new strategies in the area of pediatric HIV infection in several countries. A consortium had a working group that supported researchers on several topics related to
The topic may not be original, there are certainly other consortiums and groups that work on other topics and diseases, however it is relevant and contributes to strengthening research groups, which often do not have logistical, ethical and international support. In addition to the language barrier, countries have different regulations regarding the use of clinical samples and especially when dealing with children, who in the vast majority of cases do not respond to them. Clarity and ethics in research involving human beings are not always respected. Therefore, a homogeneous support group that guides and monitors can reduce differences and inequities, including between scientific partners from countries considered rich and poor.
RESPONSE 1: Dear reviewer, thanks for the relevant considerations. We have specifically mentioned these issues both in the introduction and in the discussion.
COMMENTS 2:
This publication is very specific and reports on the construction of a specific project. Addresses the activities that were developed. The ethical issue was extensively scrutinized, addressing the possibilities and bases on which the consent form was built to allow the protection of the subject in question (child). It was reported the construction of a standard operating procedure with documents and flows in accordance with some legal and scientific rules and legislation of the countries involved. The authors explained the conditions for transporting human specimens for use in research. He reported how agreements were developed for sharing and transferring children's materials and samples. Therefore, I consider that there is an increase in relevant information with this publication, which could contribute as an initial example of building another consortium
I considered the article well written, there was clarity in the authors' statements. Perhaps include the limitations and barriers that the working group had to overcome, as well as whether there were impediments and prohibitions in continuing with the consortium in any country specifically resulting from legislation.
The conclusion was described together with the discussion and discusses the benefits of a public-private articulation to promote strengthening in the specific research area of children living with
the HIV virus and is in line with what was previously stated. He cites the benefits of this consortium for multinational research that involves different actors and which in turn requires the sharing of clinical samples for the development of research. Despite the benefits for universal research, the benefit for the private sector that participates in these consortia was not very clear and it is not clear whether there would be a financial gain for the working groups of these consortia. It also does not make clear how the length of the actions of the entire process carried out by the working group is monitored.
Few bibliographies were cited, but I understand that it was an experiment, which probably limited the number of bibliographies. Did not compromise the article.
RESPONSE 2: Thank you for this comment. This is true, few articles deal with the same topic addressed in our paper. Nevertheless, we have performed a new bibliographic search and added two additional references.
COMMENTS 3:
The table provides a contribution regarding the differences in rules and legislation between countries and also provides possible solutions. It is a self-explanatory table and I consider it interesting to evaluate because even though it is research and relevant, it is understood that the legislation of a country is sovereign and must be respected.
RESPONSE 3: Thank you, very well appreciated.
Reviewer 2 Report
Comments and Suggestions for Authors
The manuscript titled "Sharing data and transferring samples within pediatric clinical studies: how to overcome challenges and make them a science opportunity" focuses on the requirement of sharing data for the betterment of the population living in different geographical boundaries. The manuscript highlighted the barriers to sharing the data due to the ethical guidelines adopted by the countries. The article does not attract me to read. I suggest the authors to add some measurable expected outcomes in the manuscript to attract the readers.
Author Response
COMMENTS 1:
The manuscript titled "Sharing data and transferring samples within pediatric clinical studies: how to overcome challenges and make them a science opportunity" focuses on the requirement of sharing data for the betterment of the population living in different geographical boundaries. The manuscript highlighted the barriers to sharing the data due to the ethical guidelines adopted by the countries. The article does not attract me to read. I suggest the authors to add some measurable expected outcomes in the manuscript to attract the readers.
RESPONSE 1: Dear reviewer, thank you for your comment. We were challenged from your insightful and useful comment on the measurable expected outcomes end key performance indicators, and proposed some of them in the discussion.
Reviewer 3 Report
Comments and Suggestions for Authors
This Perspective by Landi et al., aims to share best practices and learnings from the multi-national EPIICAL consortium, primarily centered on topics such as establishment of stringent ethical and informed consent protocols, abidance with complex legal frameworks across many national jurisdictions, data sharing, and transfer of patient material throughout the consortium. The authors rightly stress that the nature of the individuals recruited for the study (i.e., pediatric patients) further adds an element of legal and ethical complexity that must be navigated. Overall, it is clear from this Perspective that the study designers have painstakingly worked to establish a robust and stringent ethical and legal framework to execute the study, and the information contained within will be a useful template/resource to the field, particularly when designing similar studies (the inclusion of many study documents is appreciated). While I think the manuscript is almost there, there are a few additions that would strengthen the overall narrative and elevate this paper to be a true resource for the field:
Major Points:
· The authors stress multiple times that ethics/regulatory experts were consulted during the design and execution of the study. Could you expand on this – did the study directly interact with a “point person” at a health department or health ministry within the countries in which the study was performed? How was this relationship managed, and how was effective communication with regulators maintained? This is a difficult aspect of many similar consortia, and I think the readership of this Perspective will find detail useful here.
· I think it is worth discussing some of the key success stories/key findings of this study, and how they were enriched/made possible by the careful regulatory frameworks established by the consortium. They do not have to be new/unpublished findings, but if some aspect of the study truly worked well because of these frameworks, this would be interesting to the reader.
· Similarly, providing examples of key specific challenges that arose during this study, and how they were solved or minimized by the careful design/planning of the study would also be of interest to the reader. Are there any aspects of the study framework the authors would change in the future?
· How was data shared between sites? Briefly provide some detail here, e.g., if database software was used, protections that were employed, etc.
· The table with issues/solutions is quite good and summarizes some of the key learnings in a digestible way. I think there is opportunity to expand on some of these ideas with specific examples, hence point 2 and 3 above.
Minor Points:
· There is a typo on lines 59-60, do you mean “these would result in useful tools, strategies,” etc.?
· Reword sentence on lines 166-168. I’m not really sure what this means.
·. Typo on line 189, do you mean “reach the age of maturity” instead of majority?
· Can you please explain or reword the sentence on lines 306-308. What do you mean by growing use of “multi-sources data, AI, and multi-media electronic remote tools”? One might argue some of these might assist the study in some ways. I’m not sure what this paragraph is trying to get across.
· Can you reword/re-write the future direction on lines 309-311? The one provided, i.e., a close collaboration between study team and regulatory/ethical experts provides good guidance, is, I think, the overall point of this Perspective. I would expect a future direction to be next steps/taking learnings from this study and potentially improving future study designs in a specific way.
· On the 7th row of the table, under “ethics/regulatory issue” it says, “need to rule samples transfer outside the country”. What does this mean? Should “rule” be used here?
· The row second from the bottom of the table, under “solutions” is empty. Is it the same as the one above? Should they be combined?
Author Response
COMMENTS 1: · The authors stress multiple times that ethics/regulatory experts were consulted during the design and execution of the study. Could you expand on this – did the study directly interact with a “point person” at a health department or health ministry within the countries in which the study was performed? How was this relationship managed, and how was effective communication with regulators maintained? This is a difficult aspect of many similar consortia, and I think the readership of this Perspective will find detail useful here.
RESPONSE 1: The “Working Group (WG)” was composed of ethics and regulatory experts that were part of the Consortium and the project. They interacted mainly with the Principal investigators that in turn liaised with competent Ethics Committees for the submission of study documents. All details are provided in the paragraph “Set up of the activities”. We slightly modified the text to make this clearer, thanks.
COMMENTS 2: · I think it is worth discussing some of the key success stories/key findings of this study, and how they were enriched/made possible by the careful regulatory frameworks established by the consortium. They do not have to be new/unpublished findings, but if some aspect of the study truly worked well because of these frameworks, this would be interesting to the reader.
RESPONSE 2: We mentioned success “stories/findings” of this work in Table 1 that also provides solutions implemented by the careful regulatory frameworks established by the consortium. In particular, we consider the preparation of a letter for the PIs with practical information on how to comply with GDPR one of them. This is because we shared this letter with PIs even before the full application of the ‘new rule’, i.e.GDPR. Guide was given on how to modify the informed consent and assent processes and documents. This letter was prepared by the WG and delivered to all the PIs of the Consortium. This example has been now added in the discussion part. Thank you for the suggestion. This is also mentioned in Table 1.
COMMENTS 3: · Similarly, providing examples of key specific challenges that arose during this study, and how they were solved or minimized by the careful design/planning of the study would also be of interest to the reader. Are there any aspects of the study framework the authors would change in the future?
RESPONSE 3: The challenges and the solutions were summarized in table 1. With regards to the aspects to possibly improve, considering that the ethics and regulatory activities need timely planning (e.g.., the obtainment of an ethics approval, protocol amendment, etc.), an efficient communication between the PIs and the WG is to be sought in case of new ideas or changes of the original plans. Thank you for pointing out this.
In addition, we have proposed a set of expected outcomes and performance indicators to be measured to value and monitor the work done from an ethics/regulatory WG, and to promptly identify issues and related solutions. All these considerations have been added in the discussion.
COMMENTS 4: · How was data shared between sites? Briefly provide some detail here, e.g., if database software was used, protections that were employed, etc.
RESPONSE 4: Clinical data was entered by the study team at each participating center into a centralized database provided by the Sponsor (REDCap). Each site was responsible for entering the data collected at their center, with access restricted to the dataset pertaining to their own patients.
Access to REDCap was granted only to members of the clinical site’s staff who had been trained in the study protocol and the use of the database, and who had been delegated for data entry. Access was provided via email with a username and a temporary password, which each user was required to replace before their first login, adhering to the security criteria established by the Sponsor. The password was valid for a maximum of ninety days, after which REDCap automatically prompted the user to update it. Users were unable to access the database until a new password was set. Each patient enrolled in the study was assigned an alphanumeric code based on the principle of pseudonymization.
As part of the project activities (labs, center for statistical analysis, etc), access to the database was also granted to other project partners following the same procedures outlined above. Depending on the delegated tasks, each partner was given access only to the dataset necessary to perform their specific activities as required by the study protocols.
These details were added in the manuscript (section 4 ‘Data sharing agreeement’).
COMMENTS 5: · The table with issues/solutions is quite good and summarizes some of the key learnings in a digestible way. I think there is opportunity to expand on some of these ideas with specific examples, hence point 2 and 3 above.
RESPONSE 5: We added two examples in the text, as suggested in your second comment. Thank you so much for your input.
Minor Points:
COMMENTS 6:
- There is a typo on lines 59-60, do you mean “these would result in useful tools, strategies,” etc.?
- Reword sentence on lines 166-168. I’m not really sure what this means.
- . Typo on line 189, do you mean “reach the age of maturity” instead of majority?
RESPONSE 6: thanks for flagging these typos. They have been corrected.
COMMENTS 7: · Can you please explain or reword the sentence on lines 306-308. What do you mean by growing use of “multi-sources data, AI, and multi-media electronic remote tools”? One might argue some of these might assist the study in some ways. I’m not sure what this paragraph is trying to get across.
RESPONSE 7: An example has been added to better clarify the concept.
COMMENTS 8: · Can you reword/re-write the future direction on lines 309-311? The one provided, i.e., a close collaboration between study team and regulatory/ethical experts provides good guidance, is, I think, the overall point of this Perspective. I would expect a future direction to be next steps/taking learnings from this study and potentially improving future study designs in a specific way.
RESPONSE 8: Thank you for the suggestion. Reworded.
COMMENTS 9: · On the 7th row of the table, under “ethics/regulatory issue” it says, “need to rule samples transfer outside the country”. What does this mean? Should “rule” be used here?
RESPONSE 9: Rephrased.
COMMENTS 10: · The row second from the bottom of the table, under “solutions” is empty. Is it the same as the one above? Should they be combined?
RESPONSE 10: Yes, you are right. Thank you for pointing that out. Modified accordingly.
Round 2
Reviewer 3 Report
Comments and Suggestions for Authors
The authors have addressed my comments satisfactorily and I recommend the manuscript proceed to publication. A couple minor points:
1. Lines 317-319 and 329-331 are very similarly worded (almost sounds like copy/paste). Please reword a bit.
2. This is probably a copyediting/formatting point for the journal, but the “Expected Outcomes” section from lines 361-372 is oddly formatted.
3. Line 368 – typo, should be “Number” and not “Mumber”.